# EVALUATING GENERATIVE NETWORKS USING GAUSSIAN MIXTURES OF IMAGE FEATURES

## ABSTRACT

We develop a measure for evaluating the performance of generative networks given two sets of images. A popular performance measure currently used to do this is the Fréchet Inception Distance (FID). However, FID assumes that images featurized using the penultimate layer of Inception-v3 follow a Gaussian distribution. This assumption allows FID to be easily computed, since FID uses the 2-Wasserstein distance of two Gaussian distributions fitted to the featurized images. However, we show that Inception-v3 features of the ImageNet dataset are not Gaussian; in particular, each marginal is not Gaussian. To remedy this problem, we model the featurized images using Gaussian mixture models (GMMs) and compute the 2-Wasserstein distance restricted to GMMs. We define a performance measure, which we call WaM, on two sets of images by using Inception-v3 (or another classifier) to featurize the images, estimate two GMMs, and use the restricted 2-Wasserstein distance to compare the GMMs. We experimentally show the advantages of WaM over FID, including how FID is more sensitive than WaM to image perturbations. By modelling the non-Gaussian features obtained from Inception-v3 as GMMs and using a GMM metric, we can more accurately evaluate generative network performance.

## 1 INTRODUCTION

Generative networks, such as generative adversarial networks (GANs) (Goodfellow et al., 2014a) and variational autoencoders (Kingma & Welling, 2013), model distributions implicitly by trying to learn a map from a simple distribution, such as a Gaussian, to the desired target distribution. Using generative networks, one can generate new images (Brock et al., 2018; Karras et al., 2019a;b; 2017; Kingma & Welling, 2013), superresolve images (Ledig et al., 2017; Wang et al., 2018), solve inverse problems (Bora et al., 2017), and perform a host of image-to-image translation tasks (Isola et al., 2017; Zhu et al., 2017; 2016). However, the high dimensionality of an image distribution makes it difficult to model explicitly, that is, to estimate the moments of the distribution via some parameterization. Just estimating the covariance of a distribution requires $\frac{p(p+1)}{2}$ parameters, where $p$ is the feature dimension. For this reason, modelling distributions implicitly, using transformations of simple distributions, can be useful for high dimensional data. Since the generator network is typically nonlinear, the explicit form of the generated distribution is not known. Nonetheless, these generative models allow one to sample from the learned distribution.

Because we only have access to samples from these generative networks, instead of explicit probability density functions, evaluating their performance can be difficult. As such, several ways of evaluating the quality of the samples drawn from generative networks (Borji, 2019) have been proposed, the most popular of which is the Fréchet Inception distance (FID) (Heusel et al., 2017). FID fits Gaussian distributions to features extracted from a set of a real images and a set of GAN-generated images. The features are typically extracted using the Inception-v3 classifier (Szegedy et al., 2016a). These two distributions are then compared using the 2-Wasserstein (Villani, 2009; 2003) metric. While FID has demonstrated its utility in providing a computationally efficient metric for assessing the quality of GAN-generated images, closer examination reveals that the fundamental assumption of the FID method—namely, that the underlying feature distributions are Gaussian—is invalid. A more accurate model of the underlying features will capture a more comprehensive and informative assessment of GAN quality.

In this paper, we first show that the features used to calculate FID are not Gaussian, violating the main assumption in FID (Section 3). The 2-Wasserstein metric, which FID uses, cannot be extended past Gaussians easily because it is typically computationally intractable and does not have closed formed solutions for many families of distributions. Moreover, FID is only capturing the first two moments of the feature distribution and completely ignores all information present in the higher order moments. Missing this information biases FID toward artificially low values, an undesirable property for a performance metric.

Thus, we propose using a Gaussian mixture model (GMM) (McLachlan & Peel, 2000) for the features instead, because GMMs can model more complex distributions and capture higher order moments. GMMs are estimated efficiently and there exists a Wasserstein-type metric for GMMs (Delon & Desolneux, 2020) (Section 4) which allows us to generalize FID. We use this to develop our generative model evaluation metric, WaM. We provide code for the community to use WaM at (link will be added after acceptance).

Finally, we show that WaM is not as sensitive to visually imperceptible noise as FID (Section 5). Since GMMs can capture more information than Gaussians, WaM more accurately identifies differences between sets of images and avoids the low score bias of FID. We therefore reduce the issue of FID being overly sensitive to various noise perturbations (Borji, 2019) by modelling more information in the feature distributions. We test perturbation sensitivity using additive isotropic Gaussian noise and perturbed images which specifically attempt to increase FID using backpropagation (Mathiasen & Hvilshøj, 2020b). The ability of WaM to model more information in the feature distribution makes it a better evaluation metric for generative networks.

## 2 RELATED WORK

### 2.1 WASSERSTEIN DISTANCE

There are several ways to define a distance metric between probability distributions. A popular metric from optimal transport (Villani, 2003; 2009) is the $p$-Wasserstein metric. We first are given a Polish metric space $X$ with a metric $d$. Given $p \in (0, \infty)$ and two distributions $P$ and $Q$ on $X$ with finite moments of order $p$, the $p$-Wasserstein metric is given by

$$\mathcal{W}_p(P, Q) = \left( \inf_{\gamma} \int_{X \times X} d(x, y)^p d\gamma(x, y) \right)^{\frac{1}{p}}$$

where the infimum is taken over all joint distributions $\gamma$ of $P$ and $Q$. Different values of $p$ yield different metric properties; in image processing, the 1-Wasserstein distance on discrete spaces is used and called the earth mover distance (Rubner et al., 2000). The 2-Wasserstein metric (Dowson & Landau, 1982; Olkin & Pukelsheim, 1982) is often used when comparing Gaussians since there exists a closed form solution for

$$\mathcal{W}_2\Big(\mathcal{N}(\boldsymbol{\mu}_1, \boldsymbol{\Sigma}_1), \mathcal{N}(\boldsymbol{\mu}_2, \boldsymbol{\Sigma}_2)\Big) = \|\boldsymbol{\mu}_1 - \boldsymbol{\mu}_2\|_2^2 + \text{Tr}\left( \boldsymbol{\Sigma}_1 + \boldsymbol{\Sigma}_2 - 2\left( \boldsymbol{\Sigma}_1^{\frac{1}{2}} \boldsymbol{\Sigma}_2 \boldsymbol{\Sigma}_1^{\frac{1}{2}} \right)^{\frac{1}{2}} \right), \quad (1)$$

as is used to calculate the Fréchet Inception distance.

### 2.2 FID AND VARIANTS

The Fréchet Inception distance (FID) (Heusel et al., 2017) is a performance measure typically used to evaluate generative networks. In order to compare two sets of images, $X_1$ and $X_2$, they are featurized using the penultimate layer of the Inception-v3 network to get sets of features $F_1$ and $F_2$. For ImageNet data, this reduces the dimension of the data from $3 \times 224 \times 224 = 150{,}528$ to 2048. At this point, Heusel et al. assume that these features are Gaussian and use Equation (1) to obtain a distance between them.

There are several ways that FID has been improved. One work has shown that FID is biased (Chong & Forsyth, 2020), especially when it is computed using a small number of samples. They show that FID is unbiased asymptotically and show how to estimate the asymptotic value of FID to obtain an unbiased estimate. Others have used a network different from Inception-v3 to evaluate data that is not from ImageNet; for example, a LeNet-like (LeCun et al., 1989) feature extractor can be used

for MNIST. In this work we focus on several different ImageNet feature extractors because of their widespread use. Modelling ImageNet features has been improved due to a conditional version of FID (Soloveitchik et al., 2021) which extends FID to conditional distributions, and a class-aware Fréchet distance (Liu et al., 2018) which models the classes with GMMs. In this work, we do not consider conditional versions of FID, but our work can be extended to fit such a formulation in a straightforward manner. Moreover, we use GMMs over the feature space rather than one component per class as is done in the class-aware Fréchet distance.

Another metric related to our proposed metric is called WInD (Dimitrakopoulos et al., 2020). WInD uses a combination of the 1-Wasserstein metric on discrete spaces with the 2-Wasserstein metric on $\mathbb{R}^p$. For this reason, it is not a $p$-Wasserstein metric in $\mathbb{R}^p$ or between GMMs. For example, if $P$ and $Q$ are a mixture of Dirac delta functions then the WInD distance between them becomes the 1-Wasserstein distance. However, if $P$ and $Q$ are Gaussians, then the WInD distance between them becomes the 2-Wasserstein distance. Moreover, if $P$ and $Q$ are arbitrary GMMs, the relationship between WInD and the $p$-Wasserstein metrics is not clear. This means that WInD can alternate between the 1-Wasserstein and 2-Wasserstein distance depending on the input distributions. In this paper, we focus on using a metric which closely follows the 2-Wasserstein distance as is currently done with FID.

## 2.3 MW$_2$

A closed form solution for the 2-Wasserstein distance between GMMs is not known. This is because the joint distribution between two GMMs is not necessarily a GMM. However, if we restrict ourselves to the relaxed problem of only considering joint distributions over GMMs, then the resulting 2-Wasserstein distance of this new space is known. The restricted space of GMMs is quite large since GMMs can approximate any distribution to arbitrary precision, given enough mixture components. So given two GMMs, $P$ and $Q$, we can calculate

$$\text{MW}_2^2(P, Q) = \inf_{\gamma} \int_{X \times X} d(x, y)^2 d\gamma(x, y)$$

where the infimum is over all joint distributions $\gamma$ which are also GMMs. Constraining the class of joint distributions is a relaxation that has been done before (Bion-Nadal et al., 2019) due to the difficulty of considering arbitrary joint distributions. This metric, MW$_2$, appears in a few different sources in the literature (Chen et al., 2016; 2018; 2019) and has been studied theoretically (Delon & Desolneux, 2020); recently, implementations of this quantity have emerged.[1]

The practical formulation of this problem is done as follows. Let $P = \sum_{i=1}^{K_0} \pi_i \nu_i$ and $Q = \sum_{j=1}^{K_1} \alpha_j \mu_j$ be two GMMs with Gaussians $\nu_i, \mu_j$ for $i \in \{1, \ldots, K_0\}, j \in \{1, \ldots, K_1\}$. Then, we have that

$$\text{MW}_2^2(P, Q) = \min_{\gamma} \sum_{ij} \gamma_{ij} \mathcal{W}_2^2(\nu_i, \mu_j) \tag{2}$$

where $\gamma$ is taken to be the joint distribution over the two categorical distributions $\begin{bmatrix} \pi_1 & \ldots & \pi_{K_0} \end{bmatrix}$ and $\begin{bmatrix} \alpha_1 & \ldots & \alpha_{K_1} \end{bmatrix}$; hence, $\gamma$ in this case is actually a matrix. Thus, MW$_2$ can be implemented as a discrete optimal transport plan and efficient software exists to compute this (Flamary et al., 2021).

MW$_2$ is a great candidate for modelling the distance between GMMs for several reasons; most importantly, it is an actual distance metric. Since we are restricting the joint distribution to be a GMM, we see that MW$_2$ must be greater than or equal to the 2-Wasserstein distance between two GMMs. Moreover, MW$_2$ clearly approximates the 2-Wasserstein metric; Delon & Desolneux derive bounds showing how close MW$_2$ is to $\mathcal{W}_2$. It is also computationally efficient to compute because it can be formulated as a discrete optimal transport problem, making it practical. The strong theoretical properties and computational efficiency of MW$_2$ make it a prime candidate to calculate the distance between GMMs.

---

[1]https://github.com/judelo/gmmot

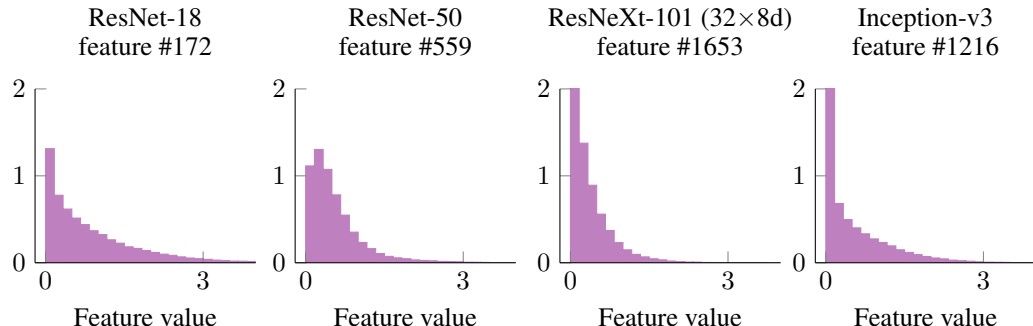

**Figure 1:** Histograms showing non-Gaussianity of randomly chosen features from the ImageNet validation dataset featurized by ResNet-18, ResNet-50, ResNeXt-101 (32×8d), and Inception-v3. They are non-negative because these features are passed through a ReLU layer and then average pooled; for this reason, we have a spike around 0. These histograms are empirical distributions and thus have an area of 1.

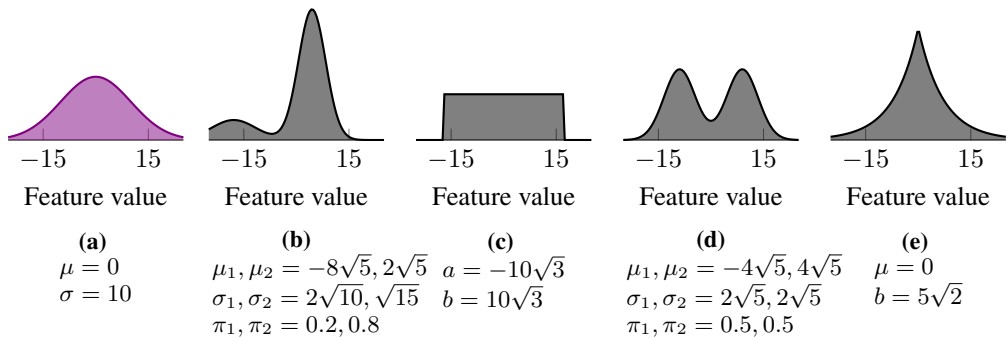

**Figure 2:** The FID score between each pair of the distributions shown above is zero although they are clearly different distributions. This is because Equation (1) is only defined for Gaussians, and FID treats any input distribution as Gaussian, even if it is not. We plot one dimensional distributions here for visualization purposes, but the FID score will remain zero even if we extend these distributions to their high dimensional isotrophic counterparts. All that is required for the FID score between two distributions to be zero is that their first two moments match. Figure 2a is the only Gaussian distribution. Figures 2b and 2d are Gaussian mixtures with two components, Figure 2c is a uniform distribution, and 2e is a Laplace distribution.

## 3 INCEPTION-V3 HAS NON-GAUSSIAN FEATURES ON IMAGENET

### 3.1 NON-GAUSSIAN FEATURES CAN DIFFER AND HAVE ZERO FID

The calculation of FID assumes that features from the penultimate layer of Inception-v3 (Szegedy et al., 2016a) are Gaussian. This layer average pools the outputs of several convolutional layers which are rectified via the ReLU activation. Though an argument can be made for why the preactivations of the convolutional layers are Gaussian (using the central limit theorem), the rectified and averaged outputs are clearly not. In fact, they are likely to be averages of rectified Gaussians (Beauchamp, 2018). Although these features are high dimensional and cannot be visualized, we plot the histograms of a randomly selected feature extracted with Inception-v3, ResNet-18, ResNet-50, and ResNeXt-101 (32×8d) in Figure 1. We construct these histograms using the 50,000 images in the ImageNet validation dataset. We see that none these randomly selected features look Gaussian.

If the Gaussian assumption of FID is false, one can achieve low FID values while having drastically different distributions, as shown on Figure 2. This is true in part because FID only considers the first two moments of the distributions being compared; differences in skew and higher order moments are not taken into account in the FID calculation. This can cause FID to be extremely low when the distributions being compared are quite different.

## 3.2 ImageNet features are not Gaussian

Testing if Inception-v3 features are Gaussian is not trivial because they are 2048-dimensional. Even if each marginal distribution appears Gaussian, we cannot be sure that the joint distribution is Gaussian. However, if the marginals are not Gaussian, this implies that original distribution is not Gaussian. Therefore, we conducted a series of Kolmogorov–Smirnov hypothesis tests (Dodge, 2008), a statistical nonparametric goodness-of-fit test that verifies whether an underlying probability distribution, in our case the marginals, differs from a hypothesized distribution, a Gaussian distribution.

We calculated features from the entire ImageNet validation dataset using ResNet-18, ResNet-50, ResNeXt-101 (32×8d), and Inception-v3. For each set of features, we then tested each marginal using the Kolmogorov–Smirnov tests with the hypothesis that the features come from a normal distribution. Using a $p$-value of 0.01, the test found that 100% of the marginals fail to pass the hypothesis. This confirms, with high certainty, that neither the marginals nor the whole feature distribution is Gaussian.

Since the features of Inception-v3 are not Gaussian, we have a few options. The first option is to use features before the average pooling layer and ReLU operation because these features may actually be Gaussian. However, these features are extremely high dimensional ($64 \times 2048 = 131{,}072$) and thus very hard to estimate accurately. Another option we have is to use a different network for feature extraction; however, most networks which perform very well on ImageNet have high dimensionality convolutional features followed by ReLU and average pooling, e.g., ResNeXt-101 (32×8d). Moreover, trying to obtain Gaussian features is not a general solution because even if the training data has Gaussian features, new data may not. Therefore, we decided to model these non-Gaussian features using Gaussian mixture models which can capture information past the first two moments of a distribution.

# 4 WaM — Model details

## 4.1 WaM — A Gaussian mixture model can learn more complex distributions

In this work we use the Gaussian mixture model (GMM) to model non-Gaussian features. GMMs are a generalization of Gaussian distributions (i.e., when the number of components equal 1) and hence we can generalize FID using the formulas discussed in Section 2.3. Moreover, any distribution can be approximated to arbitrary precision using a GMM (Delon & Desolneux, 2020). Estimation of GMM parameters are also computationally efficient and have been studied thoroughly (Bishop, 2006; McLachlan & Peel, 2000). Most importantly, we can calculate the distance between GMMs using equation 2.

We calculate our performance metric for generative models by using the MW$_2$ (Delon & Desolneux, 2020) metric for GMMs on GMMs estimated from extracted features of images. The procedure is summarized as follows: We first pick a network, such as Inception-v3, to calculate the features. These features are then used to estimate a GMM with $K$ components. We do this for real data and for generated data. We then calculate the FID of each combination of components, one from the real data GMM and one from the generated data GMM. Then, we solve a discrete optimal transport problem using the 2-Wasserstein distance squared as the ground distances to obtain WaM. We use $n = 50{,}000$ samples because this was shown to be an approximately unbiased (Chong & Forsyth, 2020) estimate of FID. We call our metric **WaM** since it is a **Wa**sserstein-type metric on GMMs of image features.

We fit the GMM to the data using the expectation maximization algorithm implemented in the scikit-learn (Pedregosa et al., 2011) package in Python. We model the features with full covariance matrices so that we are truly generalizing FID. One can fit diagonal or spherical covariance matrices if speed is required (as supported in our code), but this will yield simpler GMMs. We considered several GPU implementations of GMM fitting instead of the scikit-learn CPU implementation. However, the sequential nature of the expectation maximization algorithm caused the run times to be similar for GPU and CPU algorithms.

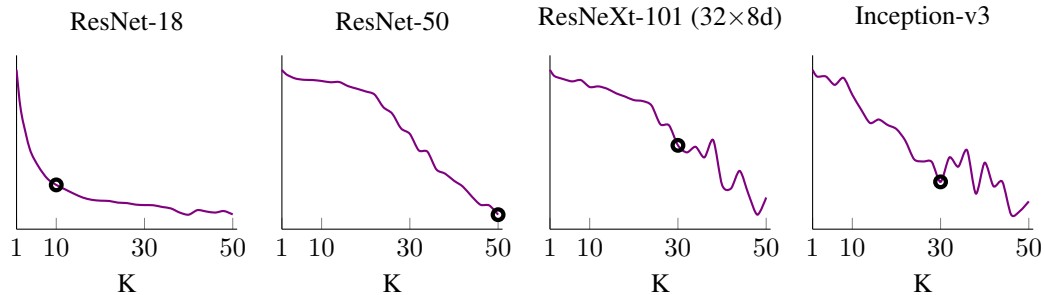

**Figure 3:** AIC curves for ResNet-18, ResNet-50, ResNeXt-101 (32×8d), and Inception-v3 features used for picking the number of mixture components $K$. We choose $K = 10$ for ResNet-18, $K = 50$ for ResNet-50, and $K = 30$ for both ResNeXt-101 (32×8d) and Inception-v3.

## 4.2 USING DIFFERENT NETWORKS

While Inception-v3 is an excellent network to use for feature extraction because of its high accuracy on the ImageNet classification task, we also use ResNet-18, ResNet-50, and ResNeXt-101 (32×8d). For each network, we use the penultimate layer for feature extraction, as was done originally for Inception-v3. We use ResNet-18 because its features are only 512-dimensional and hence can be calculated faster than Inception-v3. ResNet-50 performs better than ResNet-18 and so we included it in some of our experiments. Finally, ResNeXt-101 (32×8d) achieves the highest accuracy in the ImageNet classification task of all the pretrained classifiers on Pytorch (Paszke et al., 2019).

## 4.3 PICKING $K$ AND FITTING THE GMM

When modelling features, we must pick the number of components we choose to have in our GMM. If we pick $K = 1$ (and use Inception-v3 as our feature extractor), then we just calculate FID. The more components we pick, the better our fit will be. However, if we pick $K$ to be too large, such as $K \geq N$, then we may overfit in the sense that we can have each component centered around single data points. This is clearly not desirable, so we fit all of our GMMs with a maximum of $K = 50$ components.

We use the Akaike information criterion (AIC) to choose $K$ since likelihood criteria are well suited for density estimation (McLachlan & Peel, 2000). However, calculating AIC for multiple components will take significant computation time and power if done every time one wants to calculate WaM. For this reason, we pick a specific $K$ based on the ImageNet validation set. A value for $K$ which models the ImageNet validation dataset well should be a good $K$ for modelling similar image datasets. As shown in Figure 3, the AIC curves have varying shapes. We use the kneed method (Satopaa et al., 2011) for our choice of $K$ (using $S = 1$ in the official implementation [2]) for the ResNet-18 features since we have a textbook power-law like curve. For ResNet-50, ResNeXt-101 (32×8d), and Inception-v3 we use our best judgement since the curves do not have a knee for reasonable values of $K$.

## 5 EXPERIMENTS

### 5.1 TARGETED PERTURBATIONS — A REAL EXAMPLE OF WHEN FID FAILS AND WAM DOES NOT

Although features extracted from classifiers are not Gaussian, we do not have a perfect model for them. In fact, it is difficult to come up with distributions of features without images, because we typically have to calculate them. Hence, if we want to see when FID fails and WaM does not, we must construct data that will give us a distribution of features which cannot be modelled well with FID. This can be done if we start with a set of images, perturb them in order to increase FID, then calculate WaM on the perturbed images. Since WaM is a generalization of FID, the perturbed images will

---
[2] https://github.com/arvkevi/kneed

| Original (BigGAN) | Perturbed (BigGAN) | Original (ImageNet) | Perturbed (ImageNet) |
|---|---|---|---|
| FID = 25.13 | FID = 98.69 | FID = 2.51 | FID = 54.12 |
| WaM$^2$ = 142.12 | WaM$^2$ = 297.49 | WaM$^2$ = 52.60 | WaM$^2$ = 174.76 |

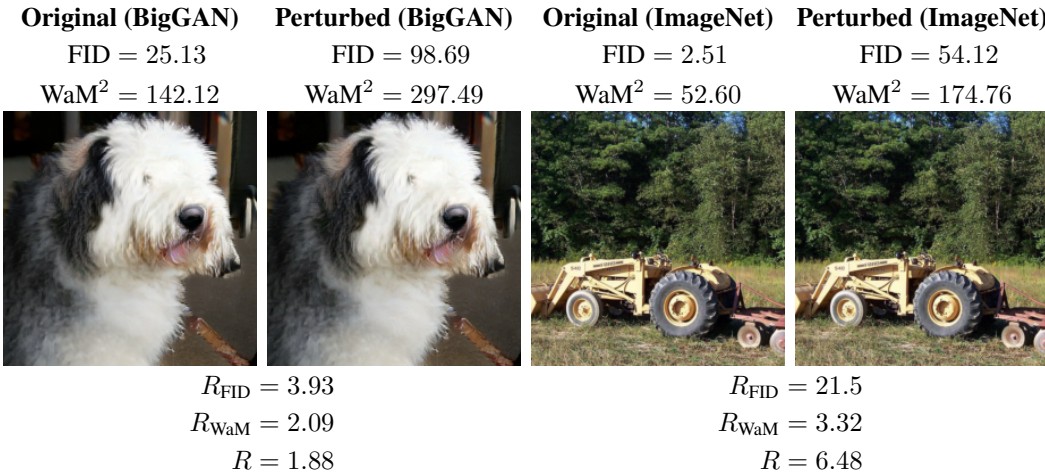

$R_{\text{FID}} = 3.93$                 $R_{\text{FID}} = 21.5$

$R_{\text{WaM}} = 2.09$                 $R_{\text{WaM}} = 3.32$

$R = 1.88$                   $R = 6.48$

**Figure 4:** Samples of images showing targeted perturbations which artificially inflate FID but not WaM. The two original images above are randomly selected from a set of 50,000 images generated by BigGAN and a set of 50,000 images of the ImageNet training dataset. We cannot visually perceive the difference between the original and perturbed images, despite the datasets from which they were selected clearly demonstrating a drastic change in FID. Note that FID of the original ImageNet training data is approximately 10 times lower than for the BigGAN generated images. The FID, WaM, and $R$ values were calculated using ResNet-18.

affect WaM as well, particularly the first and second moments of the feature distribution. However, since WaM can capture more information than FID on the feature distributions, we hypothesize that WaM will not be as affected as FID.

We construct these perturbed sets of images by backpropagating through FID using the Fast Gradient Sign Method (FGSM) (Goodfellow et al., 2014b). Backpropagation through FID was recently shown to be useful in finding adversarial examples for FID and for improving image quality of trained generators (Mathiasen & Hvilshøj, 2020b;a)[3].

To calculate FID or WaM, we must compare two sets of images; thus, we always compare to the ImageNet validation set. This allows us to calculate the FID and WaM of the ImageNet validation set against real images from the ImageNet training set, generated images from BigGAN Brock et al. (2018), and perturbed images from each. We used 50,000 images for each set when doing all the comparisons. To produce the adversarial images, we extracted the features from all the 50,000 ImageNet validation images, then ran FGSM with an $\epsilon = 0.01$ and batch size of 16 until we perturbed all 50,000 of our target images (e.g., ImageNet training set). It is worth noting that during training we calculated the gradients that maximize FID between the batch of 16 images and the extracted features the ImageNet validation set.

Comparing FID and WaM is difficult because they are different metrics with different scales. For this reason, we must normalize them when comparing. Thus, we define $R_{\text{FID}}$ to be the ratio of the FID of the perturbed images over the FID of the original images. Hence, $R_{\text{FID}}$ shows how much FID has increased due to the perturbation. Similarly, we define $R_{\text{WaM}}$ to be the ratio of WaM squared of the perturbed images over WaM squared of the original images. FID is typically reported as the 2-Wasserstein squared distance, hence we square WaM so that it is also a squared distance. Then we define $R = \frac{R_{\text{FID}}}{R_{\text{WaM}}}$ to be the ratio for these two increases. Hence, for $R > 1$ we have that FID increased faster than WaM due to perturbation.

When we perturb images generated from BigGAN (Brock et al., 2018) or the ImageNet training data we cannot visually perceive a difference, as shown in Figure 4. However, for the BigGAN images, FID increases by a factor of $R_{\text{FID}} = 3.93$ while WaM only increases by a factor of $R_{\text{WaM}} = 2.09$. This difference is even more evident with real images drawn from the ImageNet training data set. We see that the FID score after perturbation increases by $R_{\text{FID}} = 21.5$ times! Since WaM only increases by $R_{\text{WaM}} = 3.32$ times, we see that FID increased $R = 6.48$ times more than WaM for

---

[3]Although the authors of the paper introduced a Fast FID, we backpropagated through FID in our work.

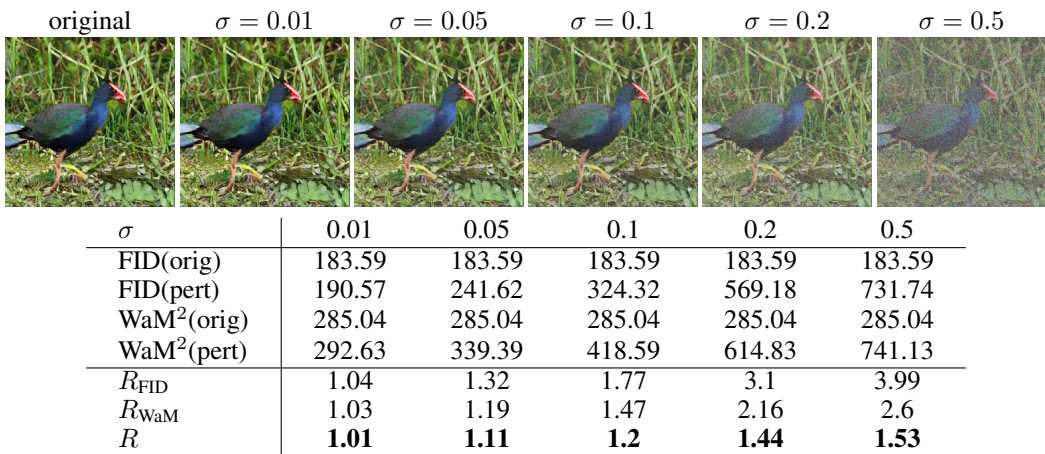

| $\sigma$ | 0.01 | 0.05 | 0.1 | 0.2 | 0.5 |
|---|---|---|---|---|---|
| FID(orig) | 183.59 | 183.59 | 183.59 | 183.59 | 183.59 |
| FID(pert) | 190.57 | 241.62 | 324.32 | 569.18 | 731.74 |
| WaM$^2$(orig) | 285.04 | 285.04 | 285.04 | 285.04 | 285.04 |
| WaM$^2$(pert) | 292.63 | 339.39 | 418.59 | 614.83 | 741.13 |
| $R_{\text{FID}}$ | 1.04 | 1.32 | 1.77 | 3.1 | 3.99 |
| $R_{\text{WaM}}$ | 1.03 | 1.19 | 1.47 | 2.16 | 2.6 |
| $R$ | **1.01** | **1.11** | **1.2** | **1.44** | **1.53** |

**Figure 5:** $R$ values for BigGAN-generated images using additive isotropic Gaussian noise showing that FID is slightly more sensitive than WaM to noise perturbations of generated images. The noise perturbations in this experiment are all greater in magnitude than the targeted perturbations in Section 5.1. The original image above was randomly selected from a set of 50,000 images generated by BigGAN. The FID, WaM, and $R$ values were calculated using ResNet-18.

an imperceptible, but targeted, perturbation. A metric which reflects perceptual quality perfectly would not be affected whatsoever by these perturbations. Neither FID nor WaM are perfect, but WaM's lower sensitivity to visually imperceptible perturbation is better aligned with the objective of assessing perceptual quality in images.

Even though these perturbations are targeted to specifically increase FID (and not WaM), this is still a fair comparison of these two metrics. WaM can learn a Gaussian distribution (e.g., if all the components are the same), yet FID and WaM yield different results in this experiment, implying that the features are not modeled well by FID and benefit from the additional information captured by WaM. Moreover, since the AIC decreases after $K = 1$, we know that WaM is using more information from the feature distributions. WaM uses more information than FID and is less sensitive than FID to these imperceptible perturbations.

Since Kernel Inception distance (KID) Bińkowski et al. (2018) may also be able to capture the higher order moments of the features, we ran our experiments using KID as well. Our results show that KID is significantly more affected than WaM by the perturbations targeted to fool FID. The details are discussed in Section A of the supplementary material. Thus, WaM is less sensitive to imperceptible FID-targeted noise than both FID and KID.

## 5.2 RANDOM PERTURBATIONS

In this section we show that WaM is also less sensitive than FID to additive isotropic Gaussian noise. We do this by corrupting images generated from BigGAN and the ImageNet training dataset by adding isotropic Gaussian noise with standard deviation $\sigma \in \{0.01, 0.05, 0.1, 0.2, 0.5\}$ and then calculating their features. Samples of how these noisy images compare to the original are shown in Figures 5 and 6. In these experiments, we use ResNet-18 to extract the features. The $\epsilon = 0.01$ used in Section 5.1 corresponds to approximately $\sigma = 0.0014$, meaning that the additive random noise in Figures 5 and 6 perturbs the images much more than the targeted noise in Figure 4.

We see that FID and WaM do not increase much when calculated using noisy BigGAN generated images, but FID skyrockets when calculated using ImageNet training data. This is likely due to FID not being able to capture the differences between the ImageNet training and validation set. One can justly assume that both data sets are sampled from the same distribution; however, we are not comparing the distributions from which they are sampled. We are comparing the two sets of images from the training and validation set, which are not the same. Therefore, FID's inability to model the correct distribution of features causes it to become extremely sensitive to this noise, even when it is barely visually perceptible. This sensitivity of FID to noise has been noted before (Heusel et al.,

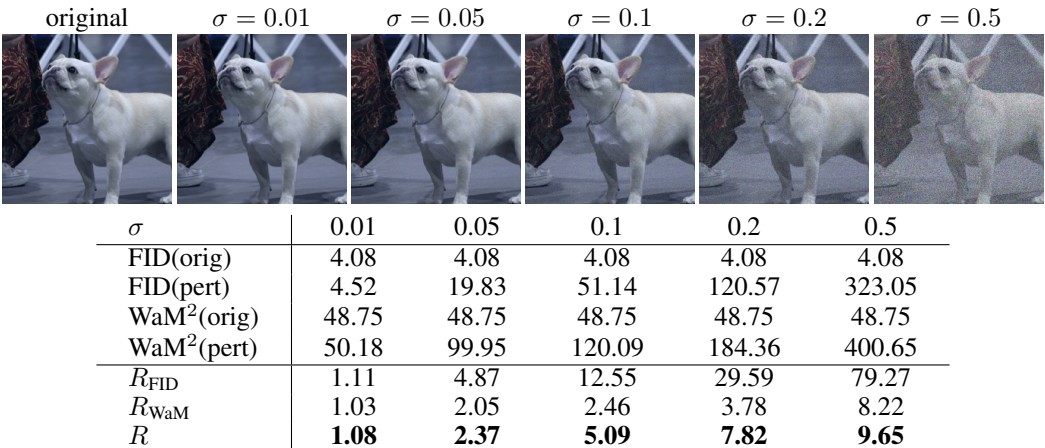

| $\sigma$ | 0.01 | 0.05 | 0.1 | 0.2 | 0.5 |
|---|---|---|---|---|---|
| FID(orig) | 4.08 | 4.08 | 4.08 | 4.08 | 4.08 |
| FID(pert) | 4.52 | 19.83 | 51.14 | 120.57 | 323.05 |
| WaM$^2$(orig) | 48.75 | 48.75 | 48.75 | 48.75 | 48.75 |
| WaM$^2$(pert) | 50.18 | 99.95 | 120.09 | 184.36 | 400.65 |
| $R_{\text{FID}}$ | 1.11 | 4.87 | 12.55 | 29.59 | 79.27 |
| $R_{\text{WaM}}$ | 1.03 | 2.05 | 2.46 | 3.78 | 8.22 |
| $R$ | **1.08** | **2.37** | **5.09** | **7.82** | **9.65** |

**Figure 6:** $R$ values for real images (ImageNet training data) using additive isotropic Gaussian noise showing that FID is significantly more sensitive than WaM to noise perturbations of real images. The noise perturbations in this experiment are all greater in magnitude than the targeted perturbations in Section 5.1. The original image above was randomly selected from a set of 50,000 images of the ImageNet training dataset. In contrast to Figure 5, we see that FID is more sensitive to these perturbations when the images look more realistic. The FID and WaM values were calculated using ResNet-18.

2017; Borji, 2019). FID is affected $R = 5.09$ times as much as WaM when the noise is barely visible ($\sigma = 0.1$), making WaM must more desirable to use in noisy contexts.

We also use KID to evaluate BigGAN-generated and ImageNet images and discuss the details in Section A of the supplementary material. Our findings show that KID has similar sensitivity as WaM on BigGAN-generated images but that KID skyrockets on ImageNet images as compared to WaM. This means that for imperceptible additive noise on realistic images, WaM is less sensitive than KID and can offer a better means to evaluate generative models which produce realistic images.

A good metric for evaluating generative network performance should be able to capture the quality of generated images at all stages. FID does not do this well. FID is sensitive to noise perturbations, especially when the images look realistic; hence, $R$ is much larger for the ImageNet training data than it is for the BigGAN generated images. As generative networks get better and better, we must use more information (not just the first and second moment) from the feature distribution in order to accurately evaluate generated samples.

## 6 CONCLUSIONS

We generalize the notion of FID by modeling image features with GMMs and computing a relaxed 2-Wasserstein distance on the distributions. Our proposed metric, WaM, allows us to accurately model more complex distributions than FID, which relies on the invalid assumption that image features follow a Gaussian distribution. Moreover, we show that WaM is less sensitive to both imperceptible targeted perturbations that modify the first two moments of the feature distribution and imperceptible additive Gaussian noise. This is important because we want a performance metric which is truly reflective of the perceptual quality of images and will not vary much when visually imperceptible noise is added. We can use WaM to evaluate networks which generate new images, superresolve images, solve inverse problems, perform image-to-image translation tasks, and more. As networks continue to evolve and generate more realistic images, WaM can provide a superior model of the feature distributions, thus enabling more accurate evaluation of extremely-realistic generated images.

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

SUPPLEMENTARY MATERIAL

## A  KERNEL INCEPTION DISTANCE EXPERIMENTS

Kernel Inception distance (KID) Bińkowski et al. (2018) is a popular method to evaluate the performance of a GAN which uses embeddings from powerful classifiers, such as Inception-v3 Szegedy et al. (2016b). We use the cubic polynomial kernel, i.e., $k(\boldsymbol{x}, \boldsymbol{y}) = (\frac{1}{d}\boldsymbol{x}^\top \boldsymbol{y} + 1)^3$ for $\boldsymbol{x}, \boldsymbol{y} \in \mathbb{R}^d$, to compute similarities between featurized samples, as is typically done. We use this method to evaluate WaM's sensitivity to imperceptible noise perturbations. To do this, we define $R_{KID}$ to be the ratio of the KID of the perturbed images over the KID of the original images. We further define $R' = \frac{R_{\text{KID}}}{R_{\text{WaM}}}$.

The targeted perturbation experiments in Section 5.1 of the original paper had values of $R = 1.88$ and $R = 6.48$ for BigGAN generated and real images, respectively. Hence, Figure 7 shows that KID is is still significantly affected by these perturbations even though they are constructed to fool FID, not KID. WaM is less sensitive than FID and KID in this setting, implying that it does not depend as heavily on the first two moments and can capture more higher order information than both metrics.

| **Original (BigGAN)** | **Perturbed (BigGAN)** | **Original (ImageNet)** | **Perturbed (ImageNet)** |
|---|---|---|---|
| KID = 0.071 | KID = 0.248 | KID = 0.006 | KID = 0.387 |
| $\text{WaM}^2 = 142.12$ | $\text{WaM}^2 = 297.49$ | $\text{WaM}^2 = 52.60$ | $\text{WaM}^2 = 174.76$ |

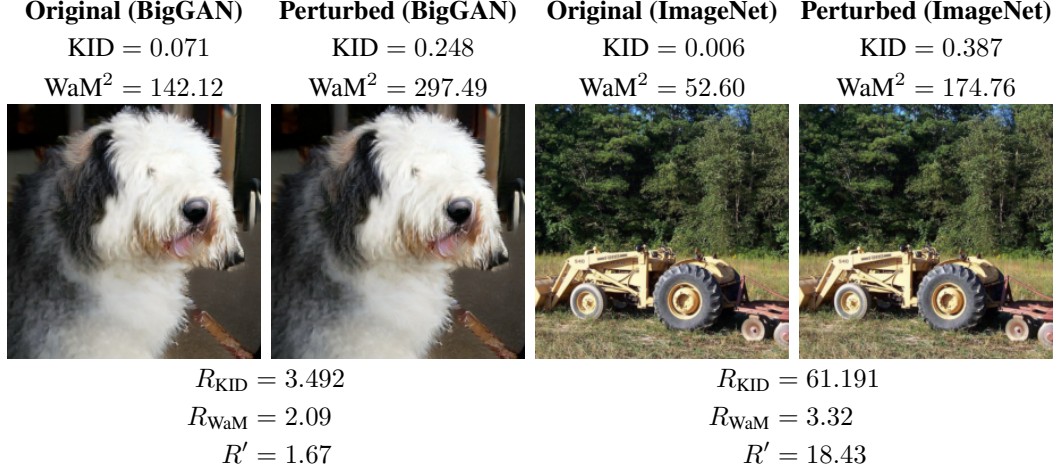

$$R_{\text{KID}} = 3.492 \qquad\qquad R_{\text{KID}} = 61.191$$
$$R_{\text{WaM}} = 2.09 \qquad\qquad R_{\text{WaM}} = 3.32$$
$$R' = 1.67 \qquad\qquad R' = 18.43$$

**Figure 7:** Samples of images showing targeted perturbations which artificially inflate FID but not WaM; however, we show KID values being indirectly inflated more than WaM. The two original images above are randomly selected from a set of 50,000 images generated by BigGAN and a set of 50,000 images of the ImageNet training dataset. We cannot visually perceive the difference between the original and perturbed images, despite the datasets from which they were selected clearly demonstrating a drastic change in KID. The KID, WaM, and $R'$ values were calculated using ResNet-18.

We now consider the random perturbations in Section 5.2 of the original paper and evaluate $R'$ on them, as shown in Figures 8 and 9. We see that KID has similar sensitivity to WaM on BigGAN generated images but much higher sensitivity on real images. In fact, KID has higher sensitivity on real images than FID. We stress that the ability to evaluate realistic images is important because that is what we **want** to generate. Therefore, WaM provides a means to evaluate realistic images better than FID and KID under imperceptible noise perturbations.

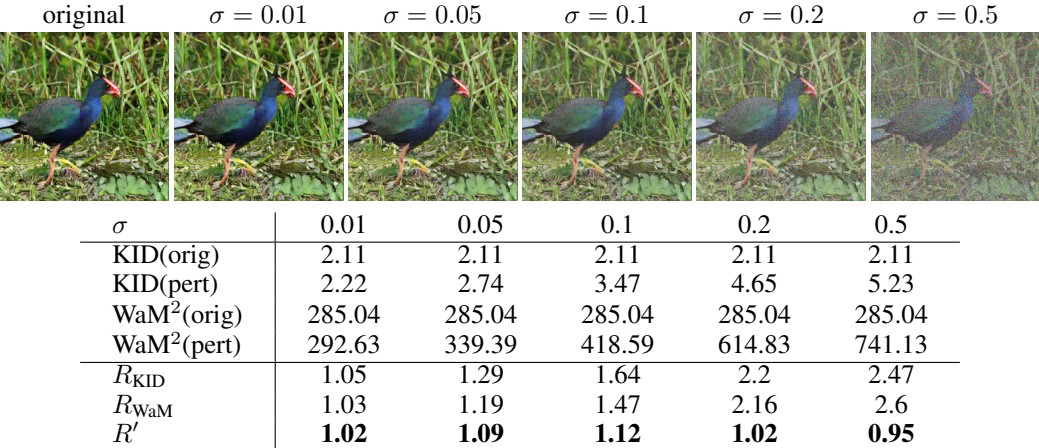

| $\sigma$ | 0.01 | 0.05 | 0.1 | 0.2 | 0.5 |
|---|---|---|---|---|---|
| KID(orig) | 2.11 | 2.11 | 2.11 | 2.11 | 2.11 |
| KID(pert) | 2.22 | 2.74 | 3.47 | 4.65 | 5.23 |
| WaM$^2$(orig) | 285.04 | 285.04 | 285.04 | 285.04 | 285.04 |
| WaM$^2$(pert) | 292.63 | 339.39 | 418.59 | 614.83 | 741.13 |
| $R_{\mathrm{KID}}$ | 1.05 | 1.29 | 1.64 | 2.2 | 2.47 |
| $R_{\mathrm{WaM}}$ | 1.03 | 1.19 | 1.47 | 2.16 | 2.6 |
| $R'$ | **1.02** | **1.09** | **1.12** | **1.02** | **0.95** |

**Figure 8:** $R'$ values for BigGAN-generated images using additive isotropic Gaussian noise showing that KID has similar sensitivity as WaM to noise perturbations of generated images. The original image above was randomly selected from a set of 50,000 images generated by BigGAN. The KID, WaM, and $R'$ values were calculated using ResNet-18.

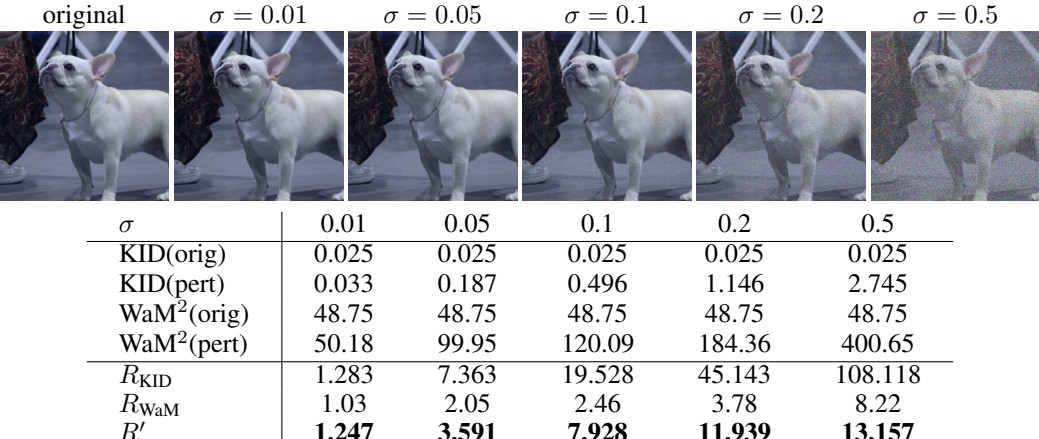

| $\sigma$ | 0.01 | 0.05 | 0.1 | 0.2 | 0.5 |
|---|---|---|---|---|---|
| KID(orig) | 0.025 | 0.025 | 0.025 | 0.025 | 0.025 |
| KID(pert) | 0.033 | 0.187 | 0.496 | 1.146 | 2.745 |
| WaM$^2$(orig) | 48.75 | 48.75 | 48.75 | 48.75 | 48.75 |
| WaM$^2$(pert) | 50.18 | 99.95 | 120.09 | 184.36 | 400.65 |
| $R_{\mathrm{KID}}$ | 1.283 | 7.363 | 19.528 | 45.143 | 108.118 |
| $R_{\mathrm{WaM}}$ | 1.03 | 2.05 | 2.46 | 3.78 | 8.22 |
| $R'$ | **1.247** | **3.591** | **7.928** | **11.939** | **13.157** |

**Figure 9:** $R'$ values for real images (ImageNet training data) using additive isotropic Gaussian noise showing that KID is significantly more sensitive than WaM to noise perturbations of real images. The original image above was randomly selected from a set of 50,000 images of the ImageNet training dataset. In contrast to Figure 5, we see that KID is more sensitive to these perturbations when the images look more realistic. The FID and WaM values were calculated using ResNet-18.

