# OpenReview forum: "Evaluating generative networks using Gaussian mixtures of image features"
_ICLR.cc/2022/Conference — ICLR 2022 Submitted_

### Official Review · Reviewer_Dqh4 · 2021-10-29

**Correctness:** 3
**Technical Novelty And Significance:** 3
**Empirical Novelty And Significance:** 3
**Recommendation:** 5
**Confidence:** 3

**Main Review:**

Strengths
1. Authors clearly indicate the shortcoming of FID, which requires the Gaussian distribution of the evaluated feature. In practical, it is hard for the extracted feature to meet this requirement. The proposed method based GMMs could overcome this issue. Directly using GMMs still faces challenge, thus authors consider the relaxed problems of only considering joint distribution over GMMs.

2. Authors express simple and convincing examples (Figures 1~2) to support the motivation. For example, I like Figure 2 to show the failure case when using FID.

3.The paper is easy to follow and well-written.

Weaknesses

1.  It seems the proposed method benefits from MW$_2$. I could not find more theoretical contribution for GAN evaluation, although them theoretical contribution is not necessary.

2. The experiment look weird except for Figures 1 and 2. The motivation of this paper is address the shortcoming of FID, which is about Gaussian assumption.  However, the experiment is to show the sensitiveness, which is not corresponding to the motivation.  Another question is about GAN evaluation,  is less sensitiveness important? as what is argued in this paper, it is true.

3. For different K,  do WaM always win compared to FID about the sensitiveness? I know it is time-consuming.

4. Do authors try to use the proposed method for more  GAN methods?  for  example,  comparing StyleGAN and StyleGANv2 methods, what is the results of the proposed method? I would like to see the application of the proposed method on GANs-based methods. Hoverer, this paper only focuses on comparison  with FID.

5. More baselines should be compared to support the proposed method, such as KID and WInD.

---------------After rebuttal-----------------------
Thanks for authors' response. I think authors did not address my concerns about why authors performance the evaluation of the sensitiveness, and more comparison with the variants of StyleGAN. I agree with $\mathbf{reviewers 3jkq}$: KID also fixes the similar issue. I change myself into negative.

**Summary Of The Paper:**

This paper propose a new method to evaluate GANs. Current prevailing  evaluation of GANs is FID, which has one assumption that the evaluated data (or feature) has Gaussian distribution. However, this assumption is false when practically applying it.  Inspired by  recent work MW$_2$,  authors propose GMMs to evaluate two sets of distribution.

**Summary Of The Review:**

Authors propose to use GMMs to evaluate GANs. I think it benefits from MW$_2$. I think the contribution is not enough. Furthermore, the experiment should be improved.

---

> ### Author Response · Authors · 2021-11-23
> **Official response to Reviewer Dqh4**
>
> Thank you for your comments and suggestions. We are happy that you found the figures useful, we tried to make them as intuitive as possible. Since you and another reviewer asked about KID, we ran our experiments again with KID and found that WaM is less sensitive to imperceptible noise than both FID and KID. See our independent post above explaining everything we have done with KID. We address a few of your other concerns below.
>
> **R values for random perterbations on BigGAN-generated images:**
>
> Since you asked if WaM outperforms FID on all values of $K$, we ran this experiment that hopefully provides enough evidence for you that WaM does outperform FID consistently. This is because WaM can capture more information than FID. See the table below; each cell is the value of R for the noise level $\sigma$ and number of components $K$ in our GMMs. Since $R > 1$ for each combination of $\sigma$ and $K$, we see that WaM beats FID consistently.
>
> |  | $\sigma = 0.01$ | $\sigma = 0.05$ | $\sigma = 0.1$ | $\sigma = 0.2$ | $\sigma = 0.5$ |
> | --- | --- | --- | --- | --- | --- |
> | $K=5$ | 1.10892076 | 1.11985634 | 1.19972936 | 1.3778139  | 1.45165715 |
> | $K=10$ |1.01126025 | 1.09861914 | 1.19287531 | 1.43198707 | 1.52504012 |
> | $K=15$ | 1.02217441 | 1.12458058  | 1.24878473 | 1.50787299 | 1.61832508 |
> | $K=20$ | 1.01022686 | 1.12114229 | 1.25485494 | 1.53802268  | 1.65563973 |
> | $K=25$ |  1.03594637 | 1.13998266 | 1.27414911 | 1.56743728 | 1.6926526 |
> | $K=30$ | 1.03646166  | 1.14185635 | 1.28691029 | 1.59029444 | 1.72108654 |
> | $K=35$ | 1.01941283 | 1.14671025 | 1.29113395 | 1.60800423 | 1.74344692 |
> | $K=40$ | 1.02859566 | 1.15585225 | 1.31097486 | 1.6329818 | 1.77296011 |
> | $K=45$ | 1.02350276 | 1.15485618 | 1.31349676 | 1.65464864 | 1.79997583 |
> | $K=50$ | 1.02780373 | 1.15735008 | 1.31972871 | 1.66776098 | 1.81876031 |
>
>
>
> **Note on sensitivity:**
>
> You are right that sensitiveity is not always important. We agree with you and it depends on the situation. What we show here is that WaM is less sensitive to FID when the perturbations *don’t visually affect the images*. That is what we care about because we don't want metrics that change drastically when the images don't visually change. If we cannot perceive the difference with our eyes, then we want our metrics to not be sensitive either. We show that WaM beats both FID and KID in this sesnse.
>
> **Note on StyleGAN and StyleGAN2:**
>
> You make a very good point that StyleGAN and StyleGAN2 are interesting models that should be evaluated. The problem with using StyleGAN and StyleGAN2 is that they are not trained on ImageNet. For that reason, FID and WaM don’t capture image quality that well since they use networks trained on ImageNet. Although you can still use FID and WaM, we perfered to stick to BigGAN as FID and WaM are better suited for it.
>
> Overall, we appreciate your comments and hope that our additional experiments are satisfying to you. Please let us know if you want us to address any additional comments that you have. Thank you.

---

### Official Review · Reviewer_xevY · 2021-11-01

**Correctness:** 4
**Technical Novelty And Significance:** 3
**Empirical Novelty And Significance:** 3
**Recommendation:** 8
**Confidence:** 4

**Main Review:**

This paper proposes an alternative metric (WaM) for the evaluation of generative networks, as an alternative to FID. The motivation behind this metric is the fact that the mathematical formulation for the FID metric requires the distributions to be Gaussian, which the authors demonstrate is not true for the Inception-v3 model. As such, the authors propose modeling both the distributions of Inception-v3 and that of the examined model as GMMs, in order to capture more information about both distributions. The authors perform two experiments to demonstrate that their proposed metric is a more robust alternative to FID, namely calculating both metrics between the ImageNet validation set and perturbed (adversarial and not) versions of either part of the ImageNet training set or a set of BigGAN generated images.

Strengths:
- The paper has a very clear structure: the authors identify a problem with the commonly used FID metric (violation of the gaussianity assumption), propose an alternative metric which is mathematically sound, and demonstrate its benefits compared to FID. This makes the paper well motivated and easy to follow.
- The motivation behind the paper is also cleanly presented. In particular, Figures 1 and 2 highlight the main problem behind the FID metric, the fact that the features of the Inception-v3 model do not follow a Gaussian distribution. This solidifies the goal behind the proposed WaM metric.
- The experiments proposed demonstrate that the WaM metric is more robust to noise in the distributions, when compared to FID. This is a useful property, since for our metric to be reliable, we expect it to not be easily affected by noise.

Negatives:
- I believe that it would be beneficial if the authors made more comparisons between WaM and other metrics used for the evaluation of GANs, other than FID. In particular, the WInD metric (Dimitrakopoulos et al. 2020), is very closely related to WaM, as the authors note (WInD also has a similar goal of circumventing the gaussianity assumption for FID). I suggest the authors further compare the theoretical aspects of WaM and WInD, as weil as calculating the WInD metric in their experiments.
- The experiments performed demonstrate that, when using BigGAN generated images, the benefit of WaM over FID is not that clear (although it is still affected less). This is especially true in the case of non-adversarial perturbations. I believe that the authors should further discuss this phenomenon, given that the overarching goal of this metric is the evaluation of generative models (currently there is some discussion at the end of section 5.2, but I believe it should be extended nonetheless).

Minor comments / Questions:
- In order to model the distributions, the number K of GMM components is identified. Is it assumed that it is the same for both distributions?
- There is a small typo in the caption of Figure 2: “isotrophic” -> “isotropic”.

References:
P Dimitrakopoulos, G Sfikas, and Christophoros Nikou. WInD: Wasserstein inception distance for evaluating generative adversarial network performance. In ICASSP 2020-2020 IEEE International Conference on Acoustics, Speech and Signal Processing (ICASSP), pp. 3182–3186. IEEE, 2020.

**Summary Of The Paper:**

This paper proposes a new metric (WaM) for the evaluation of images generated by a network, as an alternative to the commonly used FID metric. The authors show that this alternative metric is more robust to perturbations of the images (both adversarially chosen and random), when compared to FID.

**Summary Of The Review:**

I believe that this is a very clearly motivated paper, with a solid structure and experiments which support the main claims of the authors. Some comparisons with previous work on the subject are still needed in my opinion, but overall this is a solid work, and I believe it should be accepted.

---

> ### Author Response · Authors · 2021-11-23
> **Official response to Reviewer xevY**
>
> We would like to thank you for this encouraging review and we are happy that you enjoyed our paper. We were especially please that you found our figures useful because we did try to get our point accross as intuitively as possible. We have some answer for your questions below.
>
> **Theoretical difference between WaM and WInD:**
>
> The main difference between these two metrics is that WaM is a relaxation of the 2-Wasserstein distance while WInD is using the earth mover distance with FID and a ground distance. The example which shows this difference easiest is when we compare $P_1 = \mathcal N(\mu_1, \Sigma_1)$ and $P_2 = \mathcal N(\mu_2, \Sigma_2)$. We end up getting that both WaM and WInD in this case are equal to the 2-Wasserstein distance. However, now suppose that $P_1 = \sum_{i=1}^m \pi_i \delta_{x_i}$ and $P_2 = \sum_{i=1}^m \pi_i' \delta_{x_i'}$ are both sums of point masses. Then, WaM is still the 2-Wasserstein distance between the two distributions (using the 2-norm). However, WInD now becomes the 1-Wasserstein distance between the two distributions (using the 2-norm), which is called the earthmover’s distance. Therefore, since WInD changes the p in the p-Wasserstein distance, it is not a proper Wasserstein distance. The theoretical properties of WInD are not understood yet, whereas the theoretical properties of MW2 (What we use in WaM ) are explored in detail in Delon et al.
>
> **On using one value of K for each dataset:**
>
> Yes. We identify K based on the images in the validation set. We reason that if such a K is good enough to model real images, it should be good enough for generated images. This is a reasonable assumption which we find works well in practice.
>
> Thank you again for your review. We really appreciate the positive comments as well as your questions which further help us identify which aspects of our work we need to focus on.
>
> **References:**
>
> Delon, Julie, and Agnès Desolneux. "A wasserstein-type distance in the space of gaussian mixture models." SIAM Journal on Imaging Sciences 13.2 (2020): 936-970.

---

> > ### Comment · Reviewer_xevY · 2021-11-25
> > **A few more comments.**
> >
> > Thank you very much for your response.
> >
> > While some of the points I made have been covered, I am still unsure about the difference in the behavior of real images versus BigGAN generated images. In your replies to the rest of the reviewers, you mention that it is important to examine the results on real images, since as generative models get better and better, the generated ones more closely approximate real ones.
> >
> > This, however, does not fully cover the point I tried to make in my main review. More precisely, the behavior with respect to BigGAN generated images is quite different than that on real ImageNet images, which is slightly weird given that BigGAN generates quite realistic ImageNet images. I would appreciate if you could further comment on this.

---

> > > ### Author Response · Authors · 2021-11-29
> > > **Response to BigGAN-generated images**
> > >
> > > Yes you are right that the behavior of the R value is quite different between BigGAN-generated images and real images. This is because **BigGAN-generated images are lower quality than real images.** BigGAN-generated images are extremely realistic and an impressive achievement, without a doubt. However, we will discuss two ways in which BigGAN-generated images fail to model the distribution of real images.
> > >
> > > First, some classes are not modelled well at all (e.g., see 151-Chihuahua, 445-Bikini, 772-safety pin). Don’t take our word for it; we recommend trying the official tensorflow colab [here](https://colab.research.google.com/github/tensorflow/hub/blob/master/examples/colab/biggan_generation_with_tf_hub.ipynb) to generate some BigGAN images within a few minutes as we cannot upload images to openreview. This way you can see for yourself what kinds of images BigGAN can and cannot generate; plus it is a nice visualization tool.
> > >
> > > Second, BigGAN-generated images do not capture the same level of detail as real images. Some classes are modelled well and BigGAN can generate an accurate and diverse set of images. However, the details are just not there in the generated images. We recommend using the colab above to generate images from whatever class you want and compare to some real ImageNet images found on [this repo](https://github.com/EliSchwartz/imagenet-sample-images). Even though the real images on that repo are not resized to 256x256, you can still see the level of detail present. The ImageNet images are just much more detailed than the BigGAN generated images.
> > >
> > > We are not trying to bash BigGAN, because clearly it is an amazing model. However, the discrepancy in R values between BigGAN-generated images and real images comes from the inability of BigGAN to precisely model ImageNet.
> > >
> > > We thank you again for your interest in our paper and we hope that this answers your last lingering question! Take care.

---

### Official Review · Reviewer_3jkq · 2021-11-02

**Correctness:** 3
**Technical Novelty And Significance:** 1
**Empirical Novelty And Significance:** 1
**Recommendation:** 1
**Confidence:** 5

**Main Review:**

Strengths:
+ Clear writing and demonstrations.
+ Meaningful research topic.

Weaknesses:
- The technical novelty is marginal. It is a plain combination between GMM  for distribution estimation and MW2 for distance formulation.
- The technical correctness is doubtful. In Figure 1 we see the feature distributions are non-negative, single-sided, and long-tail. If a single Gaussian is not suitable enough to estimate this distribution, a mixture of Gaussians would not be much better in theory. Instead, KID [1] fundamentally handles this issue: KID does not assume a parametric form for the distribution of ReLU activation; its cubic kernel fits the long-tail distribution better; it has an unbiased estimator. Please cite [1] and discuss why the proposed WaM would be more advantageous than KID.
- The technical details are unclear. How to calculate FID when perturbing a set of images? FID is calculated by two large sets of images, say, 50k images per set. Did the authors perturb the 50k images at the same time to increase FID? If so, how if some images are perturbed much more than the others? In this case, please show the image examples with the maximal amount of perturbations. Maybe the differences are visually perceptible.
- The experiments are not convincing.
  - For the claim that FID is more sensitive than the proposed WaM, in Section 5.1 images are perturbed by backpropagating through FID. There is no wonder FID looks more sensitive given this kind of perturbations. How about the other kind of perturbations: backpropagating through WaM?
  - For Figure 5, all the FID and WaM values are in the range of 100+. The metric sensitivity in that range is meaningless because our research community cares about small FID/WaM values. BigGAN already achieves <10 FID on ImageNet. We therefore need a metric to be sensitive enough in this range to distinguish cutting-edge generative model techniques.
  - Figure 6 is meaningless neither because the results are the sensitivity on real images. Research community cares about the quality on generated images.
  - The baseline comparisons are insufficient. As mentioned above, please compare to KID [1], in terms of the sensitivity in the small value range, especially with the perturbations backpropagated through KID.

[1] Bińkowski, Mikołaj, et al. "Demystifying mmd gans." ICLR 2018.

**Summary Of The Paper:**

This paper targets to a new evaluation metric for the performance of generative models given a set of real images and a set of fake images. The authors show that Inception-v3 features of the imageNet dataset are not Gaussian and remedy this issue by modeling image features using Gaussian mixture models (GMMs) and formulating the distribution distance between two GMMs by MW2. They demonstrate their metric is less sensitive than FID against image perturbations.

**Summary Of The Review:**

See the main review.

---

> ### Author Response · Authors · 2021-11-23
> **Official response to Reviewer 3jkq**
>
> We cover the concerns that you have below.
>
> **GMMs are dense in probability spaces:**
>
> We disagree that GMMs cannot fit non-negative, single-sided, and long-tailed distributions in theory. This is provably false as GMMs are dense in the space of probability distributions and thus can approximate any distribution to arbitrary precision (Proposition 1 of Delon et al.).
>
>
> **On using KID:**
>
> Thank you for your suggestion to use KID. It was a good suggestion, which we implemented and the results are stated on an independent comment above. We are also very interested in your assertion that “KID’s cubic kernel fits the long-tail distribution better.” This would actually be very useful to understand but we have not come across this result in literature. Please provide a reference as this result would be meaningful to our research.
>
>
> **Note on technical details:**
>
> The details that you are not sure about are stated on page 7, paragraph 3:
> “To produce the adversarial images, we extracted the features from all the 50,000 ImageNet validation images, then ran FGSM with an  = 0.01 and batch size of 16 until we perturbed all 50,000 of our target images (e.g., ImageNet training set). It is worth noting that during training we calculated the gradients that maximize FID between the batch of 16 images and the extracted features of the ImageNet validation set.” In other words, for each batch of 16 images, we calculate FID between the 16 images and the 50k reference images (ImageNet validation set), perturb the 16 images using FGSM so as to maximize FID, and repeat for the next batch until we reach 50k total perturbed images. Once we have all 50k perturbed images, we calculate the true FID. We perform this perturbation only once per image using FGSM so that the maximum difference in pixel space is 0.01 (see Goodfellow et al.). Therefore, we cannot have a part of the perturbed dataset be much worse than the rest as they are all perturbed the same exact amount.
>
> **Note on the experiments:**
>
> We choose not to pursue backpropagation through WaM because that is simply not interesting. If we are able to change the higher order moments of the feature distribution but maintain the first two moments, WaM will surely be negatively affected while FID will not be. But what does this imply? Only that FID can’t capture higher moments, an undesirable property anyway. That misses the point entirely. By perturbing FID, we change the first two moments of the data, which also affects WaM. However, WaM is less affected because it can model higher order moments, a desirable property that we show in this experiment. Our aim at this experiment is showing that WaM can capture more information than FID and thus isn't as easily tricked.
>
> **Figures 5 and 6:**
>
> We disagree completely with "Figure 6 is meaningless neither because the results are the sensitivity on real images. Research community cares about the quality on generated images." We would like to emphasize that the goal of this research community is to generate images that look real. As such, an optimal GAN would generate images that are essentially the same quality as real images. Thus studying the sensitivity of FID on real images is extremely important to distinguish cutting-edge generated samples, a point that you actually make immediately before discrediting the use of real images.
>
> Although we have some disagreements, we found your suggestion to use KID very helpful and we thank you. Hopefully we have addressed your concerns by running these additional experiments and explaining our reasoning. Please let us know if you have another other concerns.
>
> **References:**
>
> Delon, Julie, and Agnès Desolneux. "A wasserstein-type distance in the space of gaussian mixture models." SIAM Journal on Imaging Sciences 13.2 (2020): 936-970.
>
> Goodfellow, Ian J., Jonathon Shlens, and Christian Szegedy. "Explaining and harnessing adversarial examples." arXiv preprint arXiv:1412.6572 (2014).

---

> > ### Comment · Reviewer_3jkq · 2021-11-30
> > **Two critical concerns are still not addressed properly**
> >
> > Thanks for the authors’ careful response! After reading the response and the other reviews, I will have to **keep my “reject” score unchanged**. This is because the following two critical concerns are still not addressed properly.
> >
> > 1. **Sensitivity on low-quality generated images (FID > 25) are meaningless.**
> > - In the original submission Figure 5, FID > 100. In the discussion window Table 1, FID > 25. The quality of these testing images is very low, which voids the meaning of FID/KID/WaM sensitivity measurements. We are looking for good metrics to differentiate top-tier generative models which can generate high-quality images with FID < 10, e.g., BigGAN on ImageNet. It is doubtful why the authors cannot achieve this level of generation in their implementation. It is further doubtful why the authors stick to the low-quality generated images to play around their sensitivity.
> >
> > 2. **It is unfair to compare the sensitivity of two metrics if the perturbation targets to fool one of them.**
> > - The proposed perturbation is optimized to fool the baseline metric FID, so that it can change FID in the steepest direction and in the most effective way. Under this perturbation, FID turns out more sensitive than the proposed WaM metric. Yet this is unfair. The reviewer suggested the authors try another perturbation that targets to fool WaM rather than FID. If this alternative perturbation showed WaM is more sensitive, the authors’ perturbation would be confirmed not neutral to compare the sensitivity of the two metrics. But the authors refused to report this experiment with the excuse “that is simply not interesting”. This is not a convincing response. Something uninteresting but necessary should still be witnessed to minimize the concern of the reviewers.
> >
> > Therefore, I encourage the authors to iterate with a resubmission with the KID experiments incorporated and my two concerns above addressed.

---

### Official Review · Reviewer_KkXg · 2021-11-04

**Correctness:** 2
**Technical Novelty And Significance:** 2
**Empirical Novelty And Significance:** 2
**Recommendation:** 3
**Confidence:** 4

**Main Review:**

Strengths: The paper has a simple yet interesting observation that the extracted features from widely-used encoder networks on widely-used image datasets are not actually distributed as Gaussian, which violates the assumption of FID. The paper then leverage recent advances on calculating 2-wasserstein distance between GMMs to obtain a generalization of FID. From this perspective, the proposed metric has the potential of addressing some limitations of FID and contributing to be a better evaluation metric for image generation.

That being said, I have a number of concerns/questions as discussed below:
- Because of the intractability of computing wasserstein distance between GMMs, the paper has to use an approximation by restricting the family of joint distribution to GMMs. In practice, how can we guarantee the quality of such approximation (which might be large)/quantify the suboptimality gap (since wasserstein between Gaussians are exact and the noise brought by approximation error may be too large wrt the signal in theory)? Moreover, as the paper discussed, this issue can be mitigated by using a large enough K (number of components). However, this may incur much more computation? For example, in equation 2, we need to compute K^2 wasserstein distance as well as a discrete optimal transport problem. When K=50 as used in the paper, we need to compute 2500x 2-wasserstein? Also a too large K will cause over-fitting, so we need to carefully tune this parameter, which is not good for a standard/widely-used evaluation metric since we need to make sure all researchers in this domain are using the same hyperparameter for fair comparisons while the optimal one is hard to know.
- At the beginning of sec 3.1, we know that preactivations of the convolution layers are Gaussian and Fig.1 indeed looks like a truncated gaussian? The truncation is mainly because the ReLU layer to force all features to be positive. Thus I may not agree with the argument in sec 3.2: "The first option is to use features before the average pooling layer and ReLU operation because these features may actually
be Gaussian. However, these features are extremely high dimensional (64  2048 = 131;072) and thus very hard to estimate accurately."
Can we only remove the ReLU operation and keep the pooling layer to reduce the dimension while the features without passing ReLU are still approximately Gaussian? It seems this option is not studied in the experiments.
- Roughly speaking, there is no new techniques proposed in this paper but a straightforward application of recent papers for calculating 2-wasserstein distance between GMMs. Thus I consider it an empirical study, where thorough and convincing experimental studies are needed. Then an important question is whether the evaluation protocols (R_FID, R_WaM, R) are good enough. From my perspective, I think there are issues with them: comparing metrics with different scales are hard and using these ratios may not make sense either. Suppose metric A range from [0, 100] and metric B range from [0, 10], also assume the sample quality linearly change within their ranges. Now if metric A change from 1 to 3 (hence R_A = 3) while metric B change from 5 to 10 (hence R_B=2), then we cannot say metric A is more sensitive than metric B because within A's range [1,3] corresponds to 2% sample quality change while [5,10] corresponds to 50% sample quality change. In short, although the authors found this issue during comparing them, I do not think the solution is good enough, which corresponds to the central empirical findings of this paper.
- The way that adversarial perturbations are constructed may lead to results that are not fair for FID. We may similarly construct adversarial examples to fool WaM by focusing the change more on third and higher moments without changing first two moments too much? Also neither adversarial constructions correspond to practical settings for evaluating deep generative models. So results are not convincing enough for demonstrating the practical value and the practical benefit is not clear. It seems to me that a better evaluation metric for generative model is one that correlates better with human perception? For example, we often observe that a set of samples look quite realistic but have a bad FID score and vice versa. So experiments about the alignment of WaM score and human perception may be needed to verify if this leads to a better metric. Robustness to some noisy perturbations may not be a central or even desirable property for a good evaluation metric, since detecting such perturbation may also be desirable in some scenatio.

---------------After rebuttal----------------------- Thank you for the response on my feedback. Some of my major concerns still remain (e.g. the justification of using ratios when comparing different metrics with different or unbounded scales, the fairness of the empirical evaluation still seems not convincing enough to me). Thus I tend to keep my original rating.

**Summary Of The Paper:**

This paper generalizes the widely-used FID metric for image generation evaluation by fitting a mixture of Gaussians instead of a single Gaussian on the extracted features. The advantage of the proposed approach is it removes the unrealistic assumption of FID that the extracted features from some encoder networks (such as Inception-v3) are approximately Gaussian. The consequence of changing to GMM is that calculating the 2-wasserstein distance now requires a relaxation/approximation, which may result in approximation error and more computation. Empirically, the paper demonstrate that the proposed metric WaM may be less sensitive to noise perturbations.

**Summary Of The Review:**

This paper propose a simple yet interesting generalization to FID by leveraging recent advances on approximating wasserstein distance between GMMs. However, as a purely empirical paper, I think there are some major flaws/issues in the evaluation procedure and thus the empirical study is not convincing enough to demonstrate the pratical advantage brought by the proposed approach.

---

> ### Author Response · Authors · 2021-11-20
> **Official response to Reviewer KkXg**
>
> We thank you for the detailed response and insightful suggestions. We cover the concerns that you have but appreciate all the positive feedback as well.
>
> **Concerns with MW2:**
>
> You are right that the MW2 distance that we use is an approximation, more specifically a relaxation, of the 2-Wasserstein distance between GMMs. Delon et al. gave bounds for this approximation and we agree that it is important to understand to what extent the bound saturates the original 2-Wasserstein distance. However, assuming that data is Gaussian, such as in FID, is a much stronger approximation then what is done with MW2. We are not trying to construct a perfect metric, just to improve on FID. Moreover, it is also worth noting that MW2 is a metric. If we consider only probability measures of GMMs, which are dense in the space of all probability measures (Proposition 1 of Delon), then MW2 induces a metric space. This means that even though MW2 does not correspond to the 2-Wasserstein metric exactly, it is still a completely valid alternative metric and can be used to distinguish probability distributions.
>
> **Choosing K:**
>
> We agree that choosing K is important and that overfitting is not desirable. As far as estimating K goes, we use AIC which is quite standard for density estimation using GMMs (See McLachlan). Overfitting typically only happens when using AIC if K is very large compared to N. Although we could technically still overfit (if a mixture component is a delta at a data point) this is unlikely because of the nature of the EM algorithm and because all parameters are randomly initialized. You are correct that computing WaM takes more time than FID. This is expected, however, since we are capturing more information (recall that WaM reduces back to FID for K=1) and thus it makes sense that more computations are used.
>
> **Rectified Gaussians:**
>
> The hidden vector after the ReLU activation (call it z) has rectified Gaussians statistics for components instead of truncated Gaussians, from our observations. This means that after the polling layer, we have an average of rectified Gaussians (Beauchamp et al.). In other words, the ReLU controls which components of z contribute to the average. Thank you for offering your suggestion of removing the ReLU as it is a good idea and one that we considered already; however, if we simply remove the ReLUs then we are not extracting the ImageNet features used in classification and are significantly modifying the feature space by forcing it to be Gaussian. This distortion of the feature space will result in a significant loss of information and that is why we chose not to pursue this line of inquiry.
>
> **R ratio:**
>
> We agree that comparing metrics in different ranges can be challenging. The provided example uses the boundedness of the metrics A and B but FID and WaM we not bounded and so the example does not apply. Although we do agree that there might exist a better way of comparing metrics, we are not aware of any such method. We have considered other alternatives and this seems like the most reasonable one. If you know of a superior technique, let us know and we would gladly look into it.
>
> **Fairness of perturbation experiments:**
>
> We assert that the perturbation experiment is a fair comparison. This experiment shows that WaM captures more information than FID . Recall that by fooling FID, we necessarily also partially fool WaM since both metrics capture first and second moment information. The point is to model more information and this experiment shows that WaM can model more information than FID and is less sensitive to perturbations that affect only the first two moments.
>
> Hopefully this response addresses your concerns at least partially. Please let us know what other concerns you have, especially the most important ones which, when addressed, would convince you to raise your score. Thank you for taking your time reviewing our paper.
>
> **References**
>
> Delon, Julie, and Agnès Desolneux. "A wasserstein-type distance in the space of gaussian mixture models." SIAM Journal on Imaging Sciences 13.2 (2020): 936-970.
>
> McLachlan, Geoffrey J., Sharon X. Lee, and Suren I. Rathnayake. "Finite mixture models." Annual review of statistics and its application 6 (2019): 355-378.
>
> Beauchamp, Maxime. "On numerical computation for the distribution of the convolution of N independent rectified Gaussian variables." Journal de la société française de statistique 159.1 (2018): 88-111.

---

### Author Response · Authors · 2021-11-23
**On using Kernel Inception Distance (KID)**

We appreciate the suggestions to use KID and we acknowledge that we should compare to metrics that may capture the non-Gaussianity of the features as well. Thus, we ran our experiments with KID to see how well it holds up to targeted and random perturbations. Table 1 below shows KID’s behavior with respect to perturbations targeted to fool FID. We define $R_{KID} = \frac{KID(pert)}{KID(orig)}$ and $R’ = \frac{R_{KID}}{R_{WaM}}$. It turns out that KID results in an R’ value that is larger than 1 as well, similarly to R for FID, implying that WaM is less sensitive than KID and FID for imperceptible noise perturbations. This is interesting especially with the targeted experiments (Section 5.1) because the perturbations are aimed to fool FID but KID suffers even more on ImageNet data.

**Table 1 (perterbations target to fool FID on BigGAN-generated and ImageNet images):**

|  |  Original (BigGAN) | Perturbed (BigGAN) | Original (ImageNet) | Perturbed (ImageNet) |
| ----------- | ----------- | --- | --- | --- |
| FID | 25.13 | 98.69 | 2.51 | 54.12 |
| WaM$^2$ | 142.12 | 297.49 | 52.60 | 174.76 |
| R$_{FID}$ || 3.93 || 21.5 |
| R$_{WaM}$ || 2.09 || 3.32 |
| **R** || **1.88** || **6.48** |
| KID | 0.071 | 0.248 | 0.006 | 0.387 |
| R$_{KID}$ || 3.492 || 61.191 |
| **R'** || **1.67** || **18.43** |

Tables 2 and 3 below show KID’s behavior on random perturbations. We notice that on BigGAN generated images, R’ is close to 1 implying that it has similar sensitivity to WaM. However, when we consider real images, R’ is even larger than R. Hence, WaM is better than FID and KID at evaluating generative networks on very realistic data that is perturbed by inperceptible noise.

**Table 2 (random perturbations on BigGAN-generated data):**

| $\sigma$ |  0.01 | 0.05 | 0.1 | 0.2 | 0.5 |
| -------- | ------- | ------ | ---- | --- | ----- |
| $R_{FID}$  | 1.04 | 1.32 | 1.77 | 3.1 | 3.99 |
| $R_{WaM}$ | 1.03 | 1.19 | 1.47 | 2.16 | 2.6 |
| **$R$** | **1.01** | **1.11** | **1.2** | **1.44** | **1.53** |
| $R_{KID}$ | 1.05  |  1.29  |  1.64  |  2.2  |   2.47 |
| $R'$ | **1.02** | **1.09** | **1.12** | **1.02** | **0.95** |

**Table 3 (random perturbations on ImageNet data):**

| $\sigma$ |  0.01 | 0.05 | 0.1 | 0.2 | 0.5 |
| -------- | ------- | ------ | ---- | --- | ----- |
| $R_{FID}$  | 1.11 | 4.87 | 12.55 | 29.59 | 79.27 |
| $R_{WaM}$ | 1.03 | 2.05 | 2.46 | 3.78 | 8.22 |
| **$R$** | **1.08** | **2.37** | **5.09** | **7.82** | **9.65** |
| $R_{KID}$ | 1.28 | 7.36 | 19.53 | 45.14 | 108.12 |
| $R'$ | **1.25** | **3.59** | **7.93** | **11.94** | **13.16** |

In conclusion, WaM is less sensitive than both FID and KID to imperceptible perturbations. We have added these results to our submission.

---

### Decision · Program_Chairs · 2022-01-20

**Decision:**

Reject

**Comment:**

This paper proposes a new evaluating metric for assessing the quality of model-generated images, that aims to correct some of the problems with the popular FID metric. The reviewers acknowledge the importance of this problem, but do not find the empirical evaluation convincing. In particular, they highlight the following issues
* Comparing FID and the new metric on examples that are adversarially selected against FID does not provide a fair comparison.
* The methods are compared on images of bad quality (FID > 25) that are therefore not informative.
* The comparison against existing techniques is incomplete
* The reviewers raise concerns about how the comparison is done quantitatively

The reviewers are not sufficiently convinced by the author response regarding these issues. I therefore recommend not accepting the paper.